# SEMI-SUPERVISED REGRESSION WITH SKEWED DATA VIA ADVERSARIALLY FORCING THE DISTRIBUTION OF PREDICTED VALUES

## ABSTRACT

Advances in scientific fields including drug discovery or material design are accompanied by numerous trials and errors. However, generally only representative experimental results are reported. Because of this reporting bias, the distribution of labeled result data can deviate from their true distribution. A regression model can be erroneous if it is built on these skewed data. In this work, we propose a new approach to improve the accuracy of regression models that are trained using a skewed dataset. The method forces the regression outputs to follow the true distribution; the forcing algorithm regularizes the regression results while keeping the information of the training data. We assume the existence of enough unlabeled data that follow the true distribution, and that the true distribution can be roughly estimated from domain knowledge or a few samples. During training neural networks to generate a regression model, an adversarial network is used to force the distribution of predicted values to follow the estimated 'true' distribution. We evaluated the proposed approach on four real-world datasets (pLogP, Diamond, House, Elevators). In all four datasets, the proposed approach reduced the root mean squared error of the regression by around 55 percent to 75 percent compared to regression models without adjustment of the distribution.

## 1 INTRODUCTION

Advances in scientific fields including drug discovery or material design are accompanied by numerous trials and errors. However, generally only representative experimental results are chosen to be reported. As a consequence of this reporting bias, the distribution of the reported results can differ from the true distribution. For this reason, when data from the literature are used to train a regression model, predictions from the regression model may differ from the true distribution because the model is derived using biased data (Lin et al., 2002; Galar et al., 2011).

In particular, pharmaceutical development is often affected by this problem. Quantitative structure-activity relationship (QSAR), including drug-target interaction (DTI), is consistently affected by the bias in the reported experimental data, because usually the targeted range of molecular property is clearly defined (Liu et al., 2015; Chen & Zhang, 2013). When regression is performed using such skewed data, it often erroneously predicts that the target properties are satisfied. As a consequence, it is difficult to discover molecules that have the desired properties (Peng et al., 2017).

Active learning applications also have a similar problem. Many active learning methods repeat the selection of new data by applying certain criteria and retraining the surrogate model (Lookman et al., 2019; Rouet-Leduc et al., 2016; Yuan et al., 2018). During this process, the data can be skewed according to the criteria

(de Mello, 2013; Prabhu et al., 2019). However, despite this problem, few studies have tried to improve the accuracy of regression models that have been trained on skewed data.

In this work, we propose a new approach to improve the accuracy of a regression model that is trained using skewed data. We assume the presence of enough unlabeled data which follow the true distribution, and that the true distribution can be roughly estimated using domain knowledge or a few examples. We use a semi-supervised learning framework with an adversarial network to force the distribution of the regression output to resemble the assumed true distribution (Figure 1). At the same time, by sharing the front part of the regression model with the encoder of an adversarial autoencoder (AAE), the process of forcing the distribution of output values is regularized in a way that the information of the labeled data is represented stably. We created skewed datasets by selecting data that exceeded a certain threshold from four real-world datasets (pLogP, Diamond, House, Elevators), then evaluated the proposed approach using these skewed datasets. The proposed approach reduced the root mean squared error (RMSE) of the regression model derived using each of the four datasets, compared to the regression model that had been trained using only the skewed datasets. We also verified that the proposed approach is feasible even when the estimate of the true distribution is not perfect.

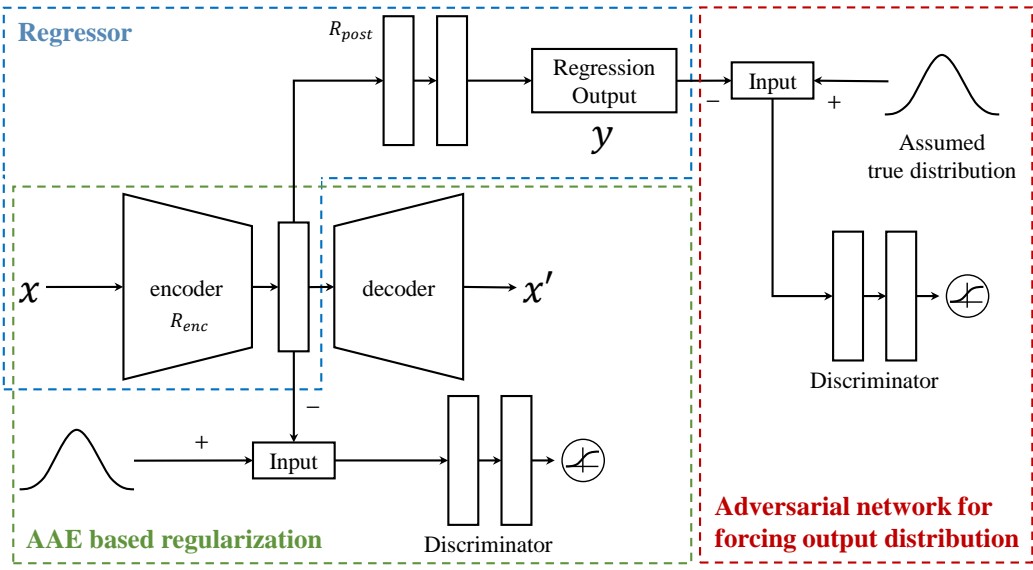

Figure 1: Architecture of a regression model with proposed approach

## 2 RELATED WORK

Semi-Supervised Learning (SSL) is a machine-learning strategy to learn using partially-labeled datasets (Chapelle et al., 2009). In the field of SSL, various methods have been developed, including those using generative models (Kingma et al., 2014), graphs (Goldberg & Zhu, 2006), self-training (Rosenberg et al., 2005) and consistency regularization (Sohn et al., 2020). SSL can improve classification and regression models by using information in a large set of unlabeled data to train a relatively small set of labeled data (Xie et al., 2019; Creswell et al., 2018; Dimokranitou, 2017; Rezagholiradeh & Haidar, 2018). These approaches generally assume that labeled and unlabeled datasets are well distributed without distortion.

Several methods were tried to train a model on data that have biased distribution. A method proposed by (Zhu et al., 2003) incorporated class prior knowledge to adjust the class distribution. Another method proposed by (Kim et al., 2020) used a model trained with skewed labeled data to generate pseudo-labels of unlabeled data for retraining by considering the entropy of prediction. Positive Unlabeled (PU) learning uses data with only positive and unlabeled samples for binary classifications (Elkan & Noto, 2008); PU learning has been studied intensively (Hsieh et al., 2019; Youngs et al., 2015). However, to the best of our knowledge, few methods used skewed labeled datasets for regression.

Although other methods for learning with imbalanced data have been developed, such as the Synthetic Minority Oversampling Technique (SMOTE) for regression (Torgo et al., 2013), these approaches can only be applied if even some of the data are in the sparse area.

## 3   METHOD

Our proposed approach is aimed to improve accuracy of regression models that are trained on a dataset that is skewed according to labels. $\mathbb{D}_l$ denotes a skewed labeled dataset with $n$ samples that satisfies

$$y_i > \theta, \qquad \forall (\boldsymbol{x}_i, y_i) \in \mathbb{D}_l = \{(\boldsymbol{x}_i, y_i)\}_{i=1}^n \tag{1}$$

where $\boldsymbol{x}_i \in \mathbb{R}^d$ is $d$-dimensional input, $y_i \in \mathbb{R}$ is ground truth target and $\theta$ is the bias threshold above which the labeled data are present. We assume that we have an estimate of the true distribution of targets $y$, $p(y)$, and that we have an unbiased unlabeled dataset $\mathbb{D}_u = \{\boldsymbol{x}_{n+1}, \boldsymbol{x}_{n+2}, \dots, \boldsymbol{x}_m\}$ that has sufficiently large $m \gg n$ and in which the distribution of ground truth targets follows $p(y)$.

In this section, we describe our approach to make the regression outputs follow the true distribution as closely as possible by using $\mathbb{D}_l$ and $\mathbb{D}_u$ in semi-supervised fashion. The proposed approach consists of two parts: "Adversarial network for forcing output distribution" and "Regularization using AAE". In the first part, we present a method that uses an adversarial network to force the output distribution of the regression model to be similar to $p(y)$. Then, we describe a method to properly regularize this forcing process by implementing an AAE.

### 3.1   ADVERSARIAL NETWORK FOR FORCING OUTPUT DISTRIBUTION

Let $\hat{y}$ be the output of a regressor that aims to predict ground truth targets $y$, and $q(\hat{y}|\boldsymbol{x})$ be the regression distribution. Then for given data distribution $p_d(\boldsymbol{x})$, a distribution of predicted regressor output values $q(\hat{y})$ can be defined as:

$$q(\hat{y}) = \int_{\boldsymbol{x}} q(\hat{y}|\boldsymbol{x}) p_d(\boldsymbol{x}) d\boldsymbol{x} \tag{2}$$

We use $\hat{y}_u$ to denote the predicted values for $\mathbb{D}_u$, and $\hat{y}_l$ to denote the predicted values for $\mathbb{D}_l$. Then the distributions of predicted values are denoted as $q(\hat{y}_u)$ and $q(\hat{y}_l)$.

The proposed approach is to regularize the regressor to have $q(\hat{y}_u)$ close to $p(y)$. To do so, an adversarial network is placed at the end of the regressor (Figure 1). The model is concurrently trained on $\mathbb{D}_l$ for regression, and on $\mathbb{D}_u$ to force the regression outputs to have similar distribution to $p(y)$. The discriminator distinguishes outputs of the regressor from randomly-sampled values in $p(y)$, and the regressor learns to deceive the discriminator.

Suppose we divide $\mathbb{D}_u$ into two subsets, $\mathbb{D}_{above}$ that consists of $\boldsymbol{x}$ that have ground truth target values $> \theta$, and $\mathbb{D}_{below}$ which consist of $\boldsymbol{x}$ that have ground truth target values $< \theta$. If the regressor is well trained

on $\mathbb{D}_l$, then prediction on $\mathbb{D}_{above}$ is expected to be relatively accurate because the distributions of $\boldsymbol{x}$ of $\mathbb{D}_l$ and $\mathbb{D}_{above}$ are similar. In contrast, for $\mathbb{D}_{below}$, the regressor should predict values that have not ever been seen; therefore, we hypothesized that during adversarial training on $\mathbb{D}_u$, the prediction on $\mathbb{D}_{above}$ should stay relatively still, whereas prediction on $\mathbb{D}_{below}$ is flexible. If this hypothesis holds, then the portion of $q(\hat{y}_u)$ that is $> \theta$ should be filled with prediction outputs on $\mathbb{D}_{above}$, and prediction outputs on $\mathbb{D}_{below}$ should be $< \theta$ to deceive the discriminator. These responses result in the regressor having $q(\hat{y}_u)$ close to $p(y)$ while maintaining accurate prediction for outputs $> \theta$.

## 3.2 REGULARIZATION BASED ON AAE

Even if prediction on $\mathbb{D}_{above}$ is intact and the predicted output distribution of $\mathbb{D}_u$ fits the assumed true distribution, the prediction on $\mathbb{D}_{below}$ will still have a large error in most cases. To guide the prediction output on $\mathbb{D}_{below}$ correctly, the information from above $\theta$ must be conveyed to below $\theta$. We used the AAE to regularize the process of forcing the output distribution to achieve appropriate propagation of the information.

We set the encoder of an AAE to share the front part of the regression network (Figure 1). We denote the shared part of the regression network as $R_{enc}$ and rest of the network as $R_{post}$. $R_{enc}$ works as the encoder of the AAE and the front part of the regression network at the same time. Therefore, latent vectors of the AAE should act as a useful feature for the $R_{post}$ which functions to predict $y$ from the latent vectors. To fulfil its function, $R_{enc}$ must consider knowledge from the labeled dataset.

Also, during the process of forcing the output distribution, $R_{post}$ also transforms the given distribution of the latent vectors to $p(y)$. To be a useful feature for $R_{post}$, latent vectors should be arranged in a similar way to $p(y)$, which possesses information about how the labeled dataset is skewed. This implies that latent vectors become more useful if $R_{enc}$ is aware of the skewed distribution of the labeled dataset. To facilitate this process, we shaped the distribution of the latent vectors to have similar characteristics to $p(y)$ by adjusting the target distribution of AAE.

Consequently, we hypothesized that the regularization process of AAE can be well guided by this information from labeled data and $p(y)$. This good guidance results in well-controlled propagation of information from the portion $> \theta$ to the prediction on $\mathbb{D}_{below}$.

## 4 EXPERIMENTS

In this section, we present results of experiments on regression of artificially-skewed datasets. For a given dataset, we first randomly sampled a sufficient amount of data to construct an unlabeled train dataset. Then we divided the rest of the data arbitrarily into three datasets to form a labeled train dataset, a validation dataset and a test dataset. To induce bias into the labeled dataset fairly, only data with labels that had values more than one standard deviation above the mean of all labels were selected randomly until the labeled dataset was sufficiently large.

For the experiments, we used four datasets: moses (QSAR on calculated pLogP), diamond, house and elevators (Polykovskiy et al., 2018; Wickham, 2016; Alcala-Fdez et al., 2011). Since QSAR on small molecules suffers from skewed datasets in many practical applications, our experiments are focused on the QSAR on calculated pLogP task. Other datasets were used to demonstrate the general applicability of the proposed approach.

We focused on comparing regression accuracy of the models that were trained using our approach or without it. Further, to explore the effectiveness at which the adversarial network forced output distribution and AAE based regularization separately, we conducted ablation studies. As the proposed approach requires an estimate of the true distribution, we tested the sensitivity of the model to the quality of the estimate. In each test, the true distribution of the data was estimated to be Gaussian distribution that has a mean and standard

Table 1: Comparisons of RMSE of different regression models. Unbiased data: Trained with fully labeled unbiased dataset, Skewed data: Trained with skewed dataset, Only AAE based regularization: Trained with skewed dataset and AAE based regularization, Only forcing output distribution: Trained with skewed dataset and forcing output distribution, Proposed approach (20 samples): Proposed approach is applied with the estimated true distribution from random 20 samples, Proposed approach (Full data): Proposed approach was applied with the estimated true distribution from full data.

| Method | pLogP | Diamond | House | Elevators |
|---|---|---|---|---|
| Skewed data | 1.689±0.087 | 0.629±0.082 | 1.694±0.027 | 1.542±0.226 |
| Only AAE based regularization | 1.654±0.035 | - | - | - |
| Only forcing output distribution | 0.982±0.729 | - | - | - |
| Proposed approach (20 samples) | 0.668±0.096 | 0.207±0.031 | 0.822±0.119 | 0.780±0.253 |
| Proposed approach (Full data) | 0.494±0.011 | 0.160±0.018 | 0.727±0.042 | 0.607±0.086 |
| Unbiased data | 0.295±0.025 | 0.127±0.019 | 0.565±0.026 | 0.374±0.017 |

deviation of 1) full data or 2) 20 randomly-sampled data. We set the assumed true distribution as Gaussian distribution for experiments on all four datasets regardless of their real distributions.

These experiments should not consider the label of unlabeled dataset until the final accuracy test. Therefore, to tune hyperparameters, we only considered accuracy of the regression on labeled dataset and distribution mismatch between predicted values and the assumed true distribution.

### 4.1 EXPERIMENTS ON QSAR TASK

The agreement between ground truth and predicted pLogP were affected by the method used. The model trained using fully-labeled data accurately predicted pLogP (Figure 2a). However, the model trained with skewed data without the proposed approach, showed decrease in accuracy as the ground truth value declined below the bias threshold (Figure 2b); this result indicates that the model's extrapolation ability is not sufficient for this task. In contrast, the model trained with the proposed approach followed the data well (Figure 2c). The RMSE (Table 1) also show that about 70% of error in the model without the proposed approach is reduced using the proposed approach. This result suggests that information learned from skewed labeled data can guide the outputs of the regression model to properly fit in the assumed true distribution during the training process; otherwise, the regression model with proposed approach would show predictions that have a distribution that matches the assumed true distribution, but would not be accurate.

### 4.2 ABLATION STUDY

To address the separate influence of the adversarial network to force output distribution and AAE for regularization, we conducted an ablation study that excluded each component.

When only the forcing output distribution part was excluded, the model showed almost uniform values outside of the labeled region (Figure 6a). The result indicates that regularization alone is not sufficient for extrapolation in this task. Comparison with (Figure 2b) demonstrates that the regularization of the predicted values reduces the variation, but that the effect of skewed dataset holds the predicted values to near the lowest labeled data point. As a result, the RMSE of the model is not meaningfully improved from that of the model without the proposed approach (Table 1).

When only AAE for regularization was excluded, the prediction values were relatively accurate but the trend of prediction according to ground truth fluctuated slightly (Figure 6b). The fluctuation suggests a decrease in the ability to convey information from labeled data. Although the RMSE of the regression model was

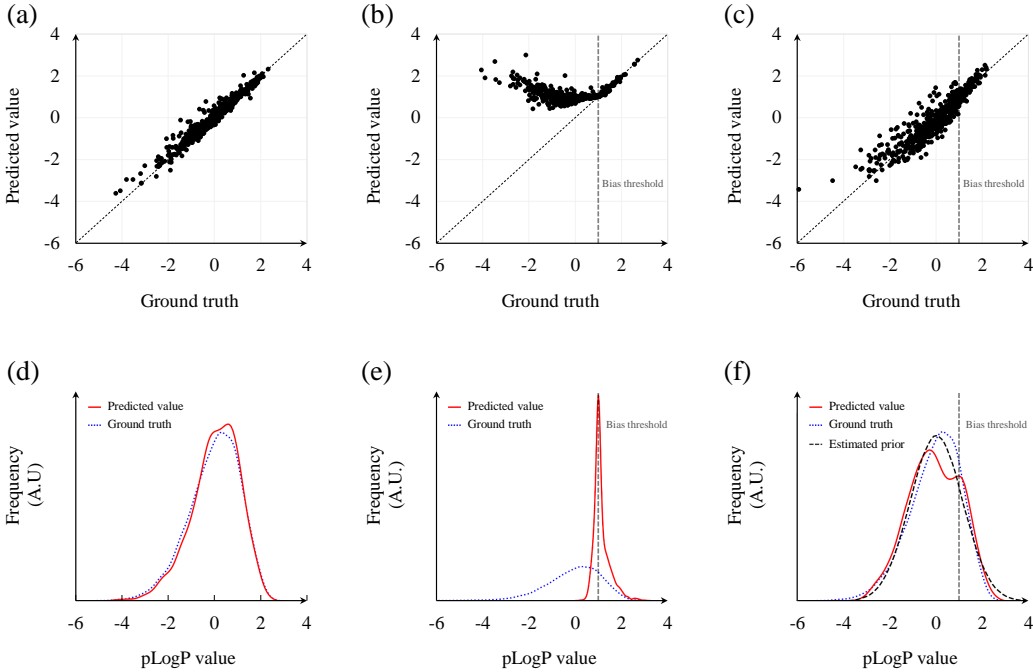

Figure 2: Correlation plot of ground truth and predicted values of regression model (a) trained on unbiased fully labeled data, (b) trained on skewed dataset and (c) trained on skewed dataset with proposed approach. Vertical dotted line: bias threshold of the dataset. (d-f) histogram of ground truth and predicted values of (a-c) respectively. Dark dashed line in (f): assumed true distribution for comparison.

$\sim 40\%$ lower than that of the model without the proposed approach (Table 1), strong regularization seems wise, because in practice, we cannot evaluate the error of the model if fully-labeled data are not available.

### 4.3 GENERAL APPLICABILITY OF PROPOSED APPROACH

RMSE were also collected for the models for datasets other than QSAR (Table 1). For the other three datasets, RMSE was reduced by 55% to 75% compared to the RMSE of the regression model trained with the skewed dataset and without the proposed approach. The result was consistent even for the difficult task that shows high RMSE (e.g., house). This result demonstrates that even slight information from the labeled data may guide the process of forcing output distribution.

However, in practice, tasks may be performed on datasets for which the true distribution cannot be estimated accurately even when the dataset is known to be skewed. The proposed approach relies strongly on the assumed true distribution, so we tested sensitivity of the proposed approach to the assumed true distribution. For this purpose, we estimated the true distribution of the datasets by using only 20 random samples instead of the full data. Interestingly, the degree of RMSE improvement over the model without the proposed approach was reduced by only 10% to 20%. Considering that the true distributions of the datasets are not exactly Gaussian, this result indicates that the proposed approach is not sensitive to error in the assumed true distribution, and is appropriate for practical applications.

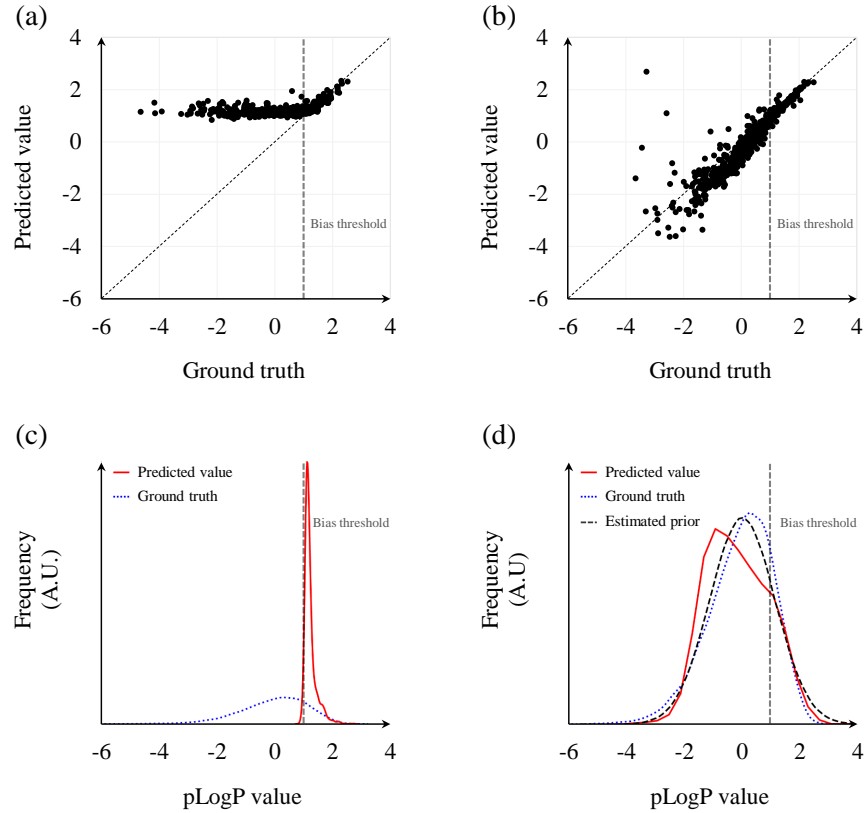

Figure 3: Correlation plot of ground truth and predicted value of regression model (a) trained on skewed dataset with only AAE based regularization and (b) trained on skewed dataset with only forcing output distribution. Vertical dotted line: bias threshold of the dataset. (c) and (d) represents histogram of ground truth and predicted values of (a) and (b) respectively. Dark dashed line: assumed true distribution for comparison.

## 5  CONCLUSION

We proposed a new approach to improve a regression model that is trained on skewed dataset. The method uses adversarial forcing to make the output distribution follow the assumed true distribution. The adversarial network to force the output distribution restrains a regression model to have the same distribution as the output distribution, and the AAE regularizes the model in the appropriate way. We evaluated the proposed approach on four datasets. The proposed approach increased the accuracy of the regression model on all datasets. Further assessment showed that the regression quality is maintained even when the proposed approach is applied with rough estimation of the true distribution.

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

# A APPENDIX

## A.1 EXPERIMENTS ON OTHER TYPES OF BIASES

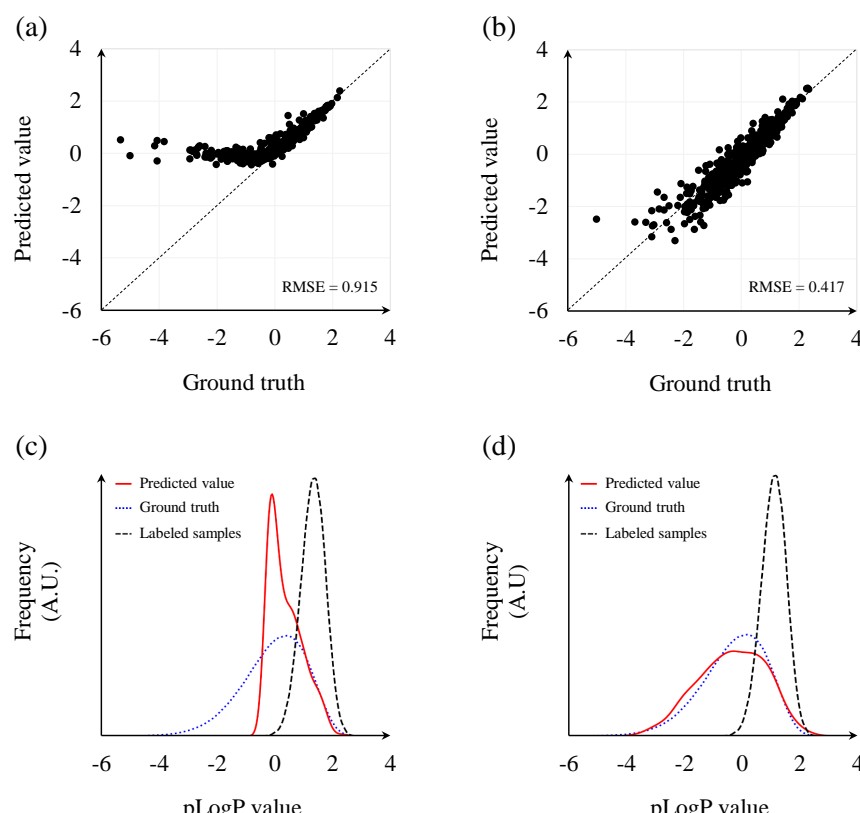

Figure 4: Correlation plot of ground truth and predicted values of regression model (a) trained on skewed dataset which have a gaussian distribution and (b) trained on the same skewed dataset with proposed approach. (c, d) histogram of ground truth and predicted values of (a, b) respectively. Dark dashed line: histogram of labeled samples in the skewed dataset.

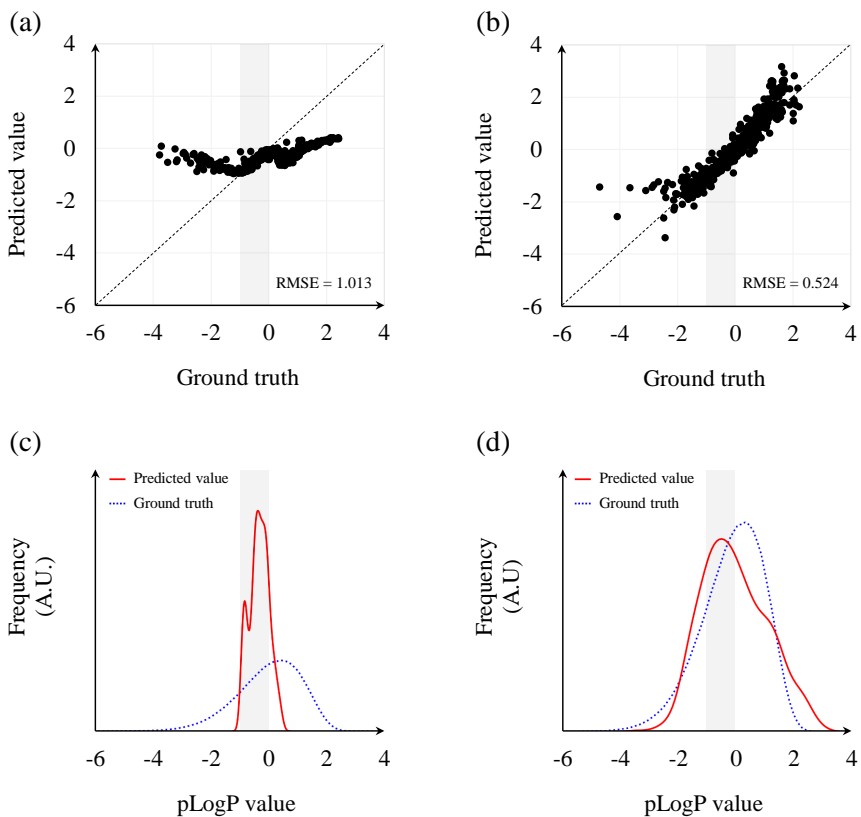

Figure 5: Correlation plot of ground truth and predicted values of regression model (a) trained on skewed dataset and (b) trained on skewed dataset with proposed approach. Labeled samples only have the ground truth values in the grey region. (c, d) histogram of ground truth and predicted values of (a, b) respectively.

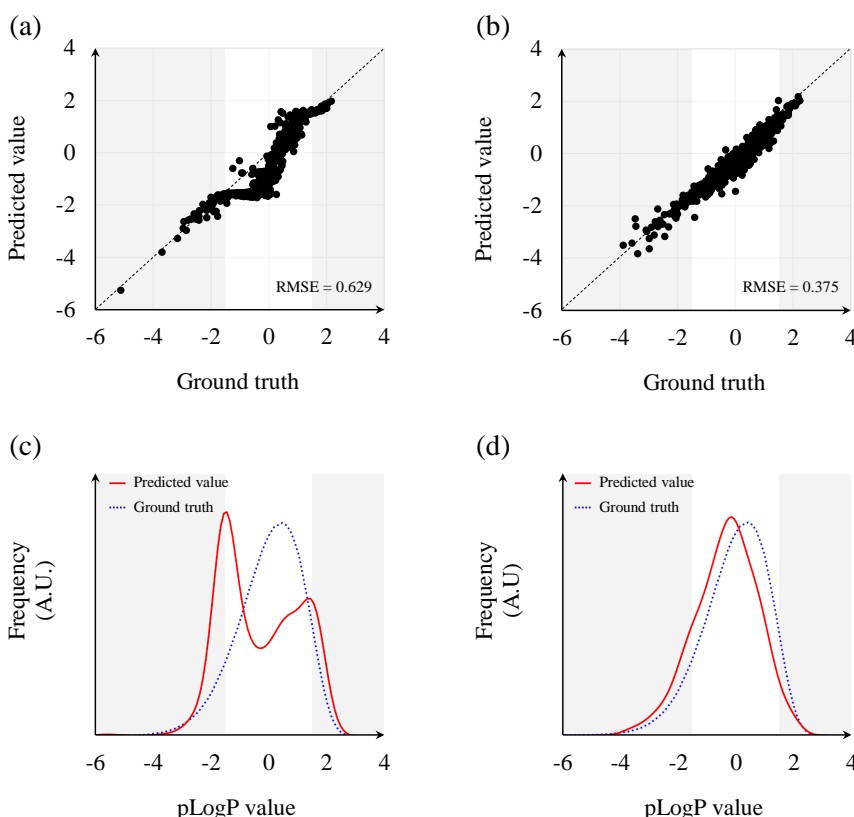

Figure 6: Correlation plot of ground truth and predicted values of regression model (a) trained on skewed dataset and (b) trained on skewed dataset with proposed approach. Labeled samples only have the ground truth values in the grey region. (c, d) histogram of ground truth and predicted values of (a, b) respectively.

