# OpenReview forum: "Semi-supervised regression with skewed data via adversarially forcing the distribution of predicted values"
_ICLR.cc/2021/Conference — Reject_

### Official Review · AnonReviewer3 · 2020-10-28

**Rating:** 6
**Confidence:** 3

**Review:**

Premise and Contributions:
This work presents a machine learning scenario where the training data is biased due to collection artifacts. The training data does not represent the true real world distribution. In order to overcome this obstacle, the work presents the following contributions:
- A semi-supervised method that uses unlabeled data (that follows the real distribution) in order to improve predictions.
- This method contains an adversarial network that forces the regression output distribution to be similar to the assumed true label distribution.
- This method contains an adversarial autoencoder that propagates the information from the assumed true distribution to the latent vector of the regression model.

Strengths:
- The scenario seems plausible.
- The proposed method seems sound. Although I have a concern with the AAE regularization.
- The experiments are convincing in my opinion, but there are some issues that I talk about below.
- The paper is pretty clear. Although, I believe the scenario is not explained in a way that is simple enough to understand instantly. But of course after reading the paper it becomes clear. Maybe there could be some improvements in the abstract and the introduction. Also I noticed that the introduction repeats the abstract, there could be some more precise explanations instead of this. For example (Kim et al. 2020)?

Issues:
- Section 3.2: "To be a useful feature for Rpost, latent vectors should be arranged in a similar way to p(y), which possesses information about how the labeled dataset is skewed." - p(y) is a distribution of labels right? How can the latent vectors have same distribution as a distribution of labels. I do not understand this.
- Is there generality with respect to different types of bias for this method? Not just biasing the dataset with labels above $\theta$
- How is the assumed true label distribution selected? What happens when it is very different from the true label distribution? Very interesting experiments could be undertaken here.
- In Section 3.1: "The model is concurrently trained on $D_l$ for regression, and on $D_u$ to force the regression outputs to have similar distribution to p(y)" - How is the network "trained" on $D_u$ is there are no labels? Or is it just running a forward pass?
- A real world application on a dataset that actually presents the issues of this specific scenario would make the value of the contributions much more convincing, instead of only testing on synthetic datasets. But I do not know about the availability of such datasets.
- Experiments: For the ablation study, how many samples are used for the "Only AAE regularization" and "Only forcing output distribution"?

My biggest concern, and a question to other reviewers and the authors: Are there any other comparable methods to the proposed method, or other methods that solve this task in comparable scenarios? Should any other method be included in the experiments section for comparison purposes?

Current decision: 5 - until I read other reviews and understand some of the moving parts better.

UPDATE:

After reading the author's updated paper and comments I have decided to improve my rating to a 6 since some of my concerns have been eased. The most important concern being eased was the type of bias was too constrained at first and now there are experiments with a more unconstrained version of bias that is more convincing.

Overall, I would say that future versions of the paper could look into a task and dataset that are close to their domain of applicability and where they can contribute an increase in performance. That would strengthen the case of this paper. I think a rating of 6 is fair for this version of the paper and I thank the authors for their efforts in updating the work and addressing my concerns directly and efficiently.

---

> ### Author Response · Authors · 2020-11-12
> **Answers for the comments**
>
> Thanks for your time in reviewing and thanks for your advice.
>
>
> 1. p(y) is a distribution of labels right? How can the latent vectors have same distribution as a distribution of labels. I do not understand this.
>
> We wanted to convey that AAE's target distribution for latent vectors would be good if you use a similar kind of distribution, not the same distribution. As mentioned in the paper, the latent distribution of AAE is transformed into distribution of the regression target through subsequent layers. The latent distribution of AAE should be any distribution that can facilitate this process.
> For example, if it is determined that the distribution of the regression target will have a gaussian distribution, it will be sufficient to use the standard gaussian distribution with a mean of 0 and std of 1 as the target distribution of AAE. Actually, in this paper, this gaussian distribution was used as the target distribution of AAE, and good performance was confirmed.
> The paper will be revised to enable more accurate delivery of these content.
>
>
> 2. Is there generality with respect to different types of bias for this method?
>
> We believe that the proposed method has the generality with respect to different types of bias because the type of bias is not an issue for the process explained in the section 3.1 and section 3.2. For the proof of the generality, we will perform additional experiments on different types of bias and update the result as supporting information.  As soon as the experiment is over, I will reply again with the results.
>
>
> 3. How is the assumed true label distribution selected? What happens when it is very different from the true label distribution? Very interesting experiments could be undertaken here.
>
> In the paper, whole experiments were performed with gaussian distribution as the label distribution since the data are expected to follow the gaussian distribution approximately. For the real application, we might select the type of distribution using domain kwoledge or sampling. As you suggested, it would be very interesting if we select other distributions such as uniform distribution or exponential distribution. We expect that if the  target distribution is too different with the true distribution, it will cause more error. However, we also expect some robustness since the section 4.3 showed promissing results.
>
>
> 4. How is the network "trained" on  D_u is there are no labels? Or is it just running a forward pass?
>
> D_u is used to train only the AAE part, excluding the regression part(R_post). After inference through the same network, the backward process is performed using only the reconstruction loss and the loss from the AAE discriminator. (Regression loss is set to zero for unlabeled data)
> The paper will be revised to enable more accurate delivery of these content.

---

> ### Author Response · Authors · 2020-11-12
> **Answers for the comments**
>
> 5. A real world application on a dataset that actually presents the issues of this specific scenario would make the value of the contributions much more convincing
>
> (Please understand that we are giving you the same answer because Reviewer 3 and Reviewer 4 also mentioned this same issue.)
> Perhaps the field that best represents the problem addressed in this paper is the field of drug discovery. In many cases, new drug development is initially aimed at finding drugs that can strongly inhibit a specific target protein with small amounts. How well it inhibits the target protein can be obtained experimentally, which takes a few weeks per molecule. Therefore, in initial development, data is prepared based on experimental data published in papers or patents. However, most of the time, papers or patents only report "active molecules" (IC50 <10uM) with good efficacy. As in this paper, only data having a value exceeding a certain threshold can be accessed selectively. When regression is performed using this data, as shown in the paper, the number of molecules with overestimated inhibitory performance increases, which makes it difficult to find the optimal molecule.
> On the other hand, in general, molecules that can be candidates for new drugs can be freely accessed through open databases such as ZINC, and the average ​​and variances of IC50 values of these candidates can be estimated relatively easily. Researchers in this domain have years of experience of performing similar experiments on various target proteins, and through it, it is not accurate, but it is possible to guess how active these candidate molecules will be on average for new targets. Also, even if not, it can be estimated by sampling 10 to 20 samples after several months of effort, and using these samples as in the paper. Considering the drug discovery process, which usually takes more than 10 years, this may be worth the investment.
> Problems such as these are, as can be seen in Liu et al., 2015, problems that are commonly encountered in this field, and a lot of efforts are being made to solve it.
> We will modify the introduction part of the paper so that these contents are well communicated.
> As mentioned in the previous examples, the proposed technique is useful when obtaining additional data is expensive or takes a long time. This is because if the proposed method is used, the performance of regression using skewed data can be improved while minimizing additional data generation.
> In addition, for the gaussian distribution, it is known that about 30 fair samples can reliably represent the distribution. Therefore, if the true distribution approximately follows the gaussian distribution, 30 fairly sampled additional data may roughly represent the true distribution. Moreover, in section 4.3, we showed that only 20 additional samples may be effective for improving the regression performance. Considering this, there will be many other cases where the proposed approach is useful.
> And as mentioned at the answer for the Reviewer2, although experiments on real-world applications with similar situations can be performed, there will be validation issues. For example, in the field of new drug development, skewed data for the experiment can be obtained from open DB. However, in this case, data that can only be obtained through real experiments are required to evaluate the performance.
> Unfortunately, since it takes a long time to perform such an experiment, We think that performance analysis will not be easy through a two-week review period.
> However, the pLogP data used in the paper was calculated from formulas based on the actual molecules and physical properties. It is real world data except that the proposed method is not actually necessary in that we can easily identify the target value.
>
> (Hui Liu, Jianjiang Sun, Jihong Guan, Jie Zheng, and Shuigeng Zhou. Improving compound-protein interaction prediction by building up highly credible negative samples. Bioinformatics, 31:i221-i229, 2015.)
>
>
> 6. how many samples are used for the "Only AAE regularization" and "Only forcing output distribution"?
>
> I am sorry, but I  am a little confused about the meaning of ‘samples’ in the question.  We performed 5 experiments for each setting. Mean and std of the target values of the full unlabeled training set w used to estimate p(y) as gaussian distribution. Training set included 800,000 unlabeled data points and 200,000 labeled data points.
>
>
> 7. Are there any other comparable methods to the proposed method, or other methods that solve this task in comparable scenarios? Should any other method be included in the experiments section for comparison purposes?
>
> We agree that it would be great if we present results of other methods that solve this task in comparable scenarios, but with a period of search, we couldn’t find them.

---

### Official Review · AnonReviewer1 · 2020-10-29
**The assumption is too strong to be useful in practice**

**Rating:** 4
**Confidence:** 3

**Review:**

This paper proposed to learn a regression model using "skewed data", which is defined as the subset of training samples with true target above certain threshold. The model consists of two components. First, the input x was mapped to its latent space through encoder R_enc. The latent representation was further mapped to the predicted output through regressor network R_post. The predictive distribution was forced to match the true target distribution p(y) through an adversarial network. Second, the latent space representations were also forced to match the true target distribution p(y). Experimental results on synthetic benchmark data showed the proposed approach performed better than naively applying regression model on the skewed data.

However, defining the "skewed data" as the subset of training samples whose target values are above certain threshold is an overly strong assumption. In practice, it's more likely that values within certain ranges were over (or under) sampled, leading to mismatch between the empirical target distribution and the true target distribution. This strong assumption makes the proposed approach not applicable to most real-world problems.

Assuming p(y) can be reliably estimated from the training data also seemed problematic. It's not clear how p(y) was estimated from the labeled dataset, where only samples with true target above certain threshold are available.

---

> ### Author Response · Authors · 2020-11-12
> **Answers for the comments**
>
> Thanks for your time in reviewing and thanks for your advice.
>
>
> 1. However, defining the "skewed data" as the subset of training samples whose target values are above certain threshold is an overly strong assumption.
>
> Considering the examples of the new drug development field mentioned in the answers to Reviewer2, Reviewer3 and Reviewer4, you can see that the skewed data assumed in this paper may not be a very extreme case. In the case of new drug development, molecules with an IC50 value of less than 10uM are defined as active molecules, and only those molecules tend to be reported with interest. Therefore, it is highly probable that data biased in a form similar to that described in the paper will be used in actual applications.
> Similar patterns can be seen not only in the drug development field, but also in many scientific fields where the criteria for successful experiments are clear. For example, in the case of low-k material defined as a material with a k value of 4 or less (a material with a lower k value compared to SiO2), only substances with a k value of 4 or less are reported in the literature in many cases.
> In addition, the assumption that there is no data below the threshold is an assumption that makes it more difficult to solve the problem. Therefore, when it is more generous, it becomes a problem that is easier to solve. In this paper, we tested the proposed algorithm using this assumption to show that it can be applied to actual applications such as the previous examples, but we think that it is still applicable even when some data in the lower area exists. We can check the performance in this case through experimentation.
> We will add to supporting information an experiment performed by creating skewed data to have a gaussian distribution with a high value as a mean. As soon as the experiment is over, I will reply again with the results.
>
>
>
> 2. Assuming p(y) can be reliably estimated from the training data also seemed problematic.
>
> As you mentioned, additional effort will be required to estimate p(y) in a situation where we have only skewed data.
> One way is to sample a small number of data from the unlabeled data pool and actually perform an experiment to create labels and estimate p(y) from it. Although it is difficult to obtain enough data to make the regression fair through these experiments, it may be possible to achieve enough data to approximate the distribution. For example, in the case of a new drug, one point of experimental data can be measured over a few weeks of experimentation, so several months of experiment can sample about 20 data as used in this paper.
> The second method is to simply estimate this from similar previous experiments or from expert experience. For example, in the field of new drugs, experts who have conducted unrecorded, unsuccessful experiments on various proteins for many years may be able to estimate how much inhibitory performance a random molecule usually has for a particular protein. And for another example, we don't know exactly the average height of people living in a city, but we can make a rough estimate. We believe that using such estimates, we can make the regression of height relatively fair even when detailed data (e.g. diet, exercise habits, posture, etc.) are known only for tall people.
> When using these estimates, how much the model becomes unstable when the estimate for p(y) is not accurate can be an important issue. As mentioned in section 4.3 of the paper, it was confirmed that there was a high performance improvement even when the experiment was performed by estimating p(y) through only 20 samples. (20 samples: actual target value of 20 unlabeled data)
> The paper will be revised to enable more accurate delivery of these contents.

---

### Official Review · AnonReviewer2 · 2020-10-29
**Straightfoward method with several problems**

**Rating:** 5
**Confidence:** 4

**Review:**

This paper proposed a semi-supervised learning approach to improve the regression model trained on output-skewed data. The key assumption is that, though the training outputs can be skewed, it is easy to estimate the true distribution of the output. The proposed model that combines an AAE that generates the output distribution, and an adversarial model that enforces the distribution of the predicted output to resemble the true distribution of the output. On several real datasets, the ablation study shows the proposed model can improve the regression accuracy.

The paper is in general well-written. However, the proposed method is straightforward, and a few important questions need to be addressed and/or clarified.

(1)	 The paper assumes that the training data are often highly skewed (intentionally) but the true distribution of the output can be easily estimated or obtained. That seems to imply that ---  it is easy to fix the data skewness by simply collecting more labeled data. Then why should we use the proposed approach? Can you provide a concrete application where getting more labeled examples is very costly yet estimating the label distribution is easy and cheap?
(2)	The paper did not explain where the target distribution in the adversarial part of AAE comes from. R_{enc} generates the latent features, which are aimed to fool the discriminator. But which target distribution do you believe the latent features should (approximately) follow?  How does it connect to using the information of the unlabeled data or the true output distribution?
(3)	The evaluation is done by creating skewed training data from existing data. This is more like a simulation. Can you incorporate a real-world application, e.g., drug design (mentioned by the paper), to showcase the performance and usefulness of the proposed approach?

---

> ### Author Response · Authors · 2020-11-12
> **Answers for the comments**
>
> Thanks for your time in reviewing and thanks for your advice.
>
>
> 1. Can you provide a concrete application where getting more labeled examples is very costly yet estimating the label distribution is easy and cheap?
>
> (Please understand that we are giving you the same answer because Reviewer 3 and Reviewer 4 also mentioned this same issue.)
> Perhaps the field that best represents the problem addressed in this paper is the field of drug discovery. In many cases, new drug development is initially aimed at finding drugs that can strongly inhibit a specific target protein with small amounts. How well it inhibits the target protein can be obtained experimentally, which takes a few weeks per molecule. Therefore, in initial development, data is prepared based on experimental data published in papers or patents. However, most of the time, papers or patents only report "active molecules" (IC50 <10uM) with good efficacy. As in this paper, only data having a value exceeding a certain threshold can be accessed selectively. When regression is performed using this data, as shown in the paper, the number of molecules with overestimated inhibitory performance increases, which makes it difficult to find the optimal molecule.
> On the other hand, in general, molecules that can be candidates for new drugs can be freely accessed through open databases such as ZINC, and the average ​​and variances of IC50 values of these candidates can be estimated relatively easily. Researchers in this domain have years of experience of performing similar experiments on various target proteins, and through it, it is not accurate, but it is possible to guess how active these candidate molecules will be on average for new targets. Also, even if not, it can be estimated by sampling 10 to 20 samples after several months of effort, and using these samples as in the paper. Considering the drug discovery process, which usually takes more than 10 years, this may be worth the investment.
> Problems such as these are, as can be seen in Liu et al., 2015, problems that are commonly encountered in this field, and a lot of efforts are being made to solve it.
> We will modify the introduction part of the paper so that these contents are well communicated.
> As mentioned in the previous examples, the proposed technique is useful when obtaining additional data is expensive or takes a long time. This is because if the proposed method is used, the performance of regression using skewed data can be improved while minimizing additional data generation.
> In addition, for the gaussian distribution, it is known that about 30 fair samples can reliably represent the distribution. Therefore, if the true distribution approximately follows the gaussian distribution, 30 fairly sampled additional data may roughly represent the true distribution. Moreover, in section 4.3, we showed that only 20 additional samples may be effective for improving the regression performance. Considering this, there will be many other cases where the proposed approach is useful.
>
> (Hui Liu, Jianjiang Sun, Jihong Guan, Jie Zheng, and Shuigeng Zhou. Improving compound-protein interaction prediction by building up highly credible negative samples. Bioinformatics, 31:i221-i229, 2015.)
>
>
>
>  2. But which target distribution do you believe the latent features should (approximately) follow?
>
> We think it would be nice if the target distribution of AAE is of a similar type to the distribution of the final regression target. As mentioned in the paper, the latent distribution of AAE is transformed into distribution of the regression targets through subsequent layers, and it would be good if the latent vectors had a distribution that could facilitate this process.
> For example, if it is determined that the distribution of the regression target will have a gaussian distribution, it will be sufficient to use the standard gaussian distribution with a mean of 0 and std of 1 as the target distribution of AAE. Actually, in this paper, this gaussian distribution was used as the target distribution of AAE, and good performance was confirmed.
>
>
>
> 3. Can you incorporate a real-world application, e.g., drug design (mentioned by the paper), to showcase the performance and usefulness of the proposed approach?
>
> In fact, it seems that additional consideration is needed for experimenting with real-world applications. Although experiments on real-world applications with similar situations can be performed, there will be validation issues. For example, in the field of new drug development, skewed data for the experiment can be obtained from open DB. However, in this case, data that can only be obtained through real experiments are required to evaluate the performance.
> Unfortunately, since it takes a long time to perform such an experiment, We think that performance analysis will not be easy through a two-week review period.

---

### Official Review · AnonReviewer4 · 2020-10-31
**Regression method for skewed data**

**Rating:** 5
**Confidence:** 2

**Review:**

##########################################################################

Summary:

The paper presents a novel approach to improve the accuracy of regression models that are learned from a skew dataset. The proposed approach consists of two parts, namely, (i) adversarial network for forcing output distributions and (ii) regularization based on an adversarial autoencoder. Experiments suggest that the proposed approach increases the accuracy of the regression model for all the four datasets considered in the paper.


##########################################################################

Pros:

(1) In practice, the distribution of reported data could be different from the true distribution. The proposed approach is potentially useful for analyzing such data.

(2) This paper provides comprehensive experiments using four datasets. All the experiments agree that the proposed approach increases the accuracy of the regression model and outperforms some existing approaches in terms of RMSE.

(3) The paper is clearly written and the motivation for the study is well explained in the introduction. Figure 1 is helpful to understand the architecture of the proposed approach.


##########################################################################

Cons:

(1) The proposed approach is based on the assumption that labeled data are skewed and unlabeled data follow the assumed true distribution. I am not sure how realistic this assumption is. For instance, the examples discussed in Introduction appear to be the ones in which both labeled and unlabeled data are skewed. Also, in the experiments given in Section 4, all the datasets are not originally skewed, but they are artificially changed to skewed datasets. It would be convincing to provide a real example in which the assumption of the proposed approach holds.

(2) In the second paragraph of Section 4.3, it is claimed that the proposed approach relies strongly on the assumed true distribution. At the same time, it is also mentioned that the proposed approach is not sensitive to error in the assumed true distribution, and is appropriate for practical applications. These two claims seem contradictory to me. More explanations about these statements would be helpful.

(3) It would be ideal if some theoretical results are presented to support the proposed approach. For example, is it possible to provide any theoretical background to support the claim that the proposed approach is not sensitive to error in the assumed true distribution?


##########################################################################

Questions during rebuttal period:

Please address and clarify the cons above.


---

### Updates:

I thank the authors for their response. Some of my concerns are addressed. However, unfortunately, I still think the assumption of the proposed approach is too strong to have broad applications. I will keep my original score.

---

> ### Author Response · Authors · 2020-11-12
> **Answers for the comments 1**
>
> Thanks for your time in reviewing and thanks for your advice.
>
> 1. It would be convincing to provide a real example in which the assumption of the proposed approach holds.
>
> (Please understand that we are giving you the same answer because Reviewer 3 and Reviewer 4 also mentioned this same issue.)
> Perhaps the field that best represents the problem addressed in this paper is the field of drug discovery. In many cases, new drug development is initially aimed at finding drugs that can strongly inhibit a specific target protein with small amounts. How well it inhibits the target protein can be obtained experimentally, which takes a few weeks per molecule. Therefore, in initial development, data is prepared based on experimental data published in papers or patents. However, most of the time, papers or patents only report "active molecules" (IC50 <10uM) with good efficacy. As in this paper, only data having a value exceeding a certain threshold can be accessed selectively. When regression is performed using this data, as shown in the paper, the number of molecules with overestimated inhibitory performance increases, which makes it difficult to find the optimal molecule.
> On the other hand, in general, molecules that can be candidates for new drugs can be freely accessed through open databases such as ZINC, and the average ​​and variances of IC50 values of these candidates can be estimated relatively easily. Researchers in this domain have years of experience of performing similar experiments on various target proteins, and through it, it is not accurate, but it is possible to guess how active these candidate molecules will be on average for new targets. Also, even if not, it can be estimated by sampling 10 to 20 samples after several months of effort, and using these samples as in the paper. Considering the drug discovery process, which usually takes more than 10 years, this may be worth the investment.
> Problems such as these are, as can be seen in Liu et al., 2015, problems that are commonly encountered in this field, and a lot of efforts are being made to solve it.
>
> We will modify the introduction part of the paper so that these contents are well communicated.
> As mentioned in the previous examples, the proposed technique is useful when obtaining additional data is expensive or takes a long time. This is because if the proposed method is used, the performance of regression using skewed data can be improved while minimizing additional data generation.
> In addition, for the gaussian distribution, it is known that about 30 fair samples can reliably represent the distribution. Therefore, if the true distribution approximately follows the gaussian distribution, 30 fairly sampled additional data may roughly represent the true distribution. Moreover, in section 4.3, we showed that only 20 additional samples may be effective for improving the regression performance. Considering this, there will be many other cases where the proposed approach is useful.
>
> (Hui Liu, Jianjiang Sun, Jihong Guan, Jie Zheng, and Shuigeng Zhou. Improving compound-protein interaction prediction by building up highly credible negative samples. Bioinformatics, 31:i221-i229, 2015.)

---

> ### Author Response · Authors · 2020-11-12
> **Answers for the comments 2**
>
> 2. In the second paragraph of Section 4.3, it is claimed that the proposed approach relies strongly on the assumed true distribution. At the same time, it is also mentioned that the proposed approach is not sensitive to error in the assumed true distribution, and is appropriate for practical applications.
>
> As we claimed in the paper, we believe that proposed approach relies strongly on the assumed true distribution. However, we don't think it is necessarily contradictory that the assumed true distribution plays a very important role, and that it is robust to the assumed true distribution. Although the assumed true distribution provides essential information in the regression process, it is not easy to know exactly which of the information inherent in the assumed true distribution are necessary information. For example, a model may be sufficient to simply know that the given data is skewed, or, conversely, it may require all detailed information about the form of the distribution. Even if it is the former case, the importance of the assumed true distribution or the dependence of the model on the assumed true distribution does not decrease. To confirm this, we conducted an experiment as shown in section 4.3 using the assumed true distribution which is somewhat inaccurate, and from the results, it has been shown that the assumed true distribution can play a sufficient role even if the detailed information is not correct.
> In addition, we agree that the proposed method is not applicable if the target distribution is completely wrong. What is important is whether the proposed method is robust enough for us to use in a real application. From that point of view, we expressed our method as robust because we thought it would be sufficient to identify performance improvements with the target distribution estimated from 20 samples.
>
>
> 3. It would be ideal if some theoretical results are presented to support the proposed approach.
>
> We explained in Section 3.1 and Section 3.2 how the proposed model conveys information from labeled data to unlabeled regions according to the estimated target distribution. Although the process is not mathematically proven, we hope that presented explanations and experiments are informative enough for supporting our claim.
> For the robustness, we think that the point is how much of ground truth targets of unlabeled data fit in the estimated target distribution. The overlapping coefficient between two distributions may represent this. Since more theoretical development will take some period, we will try to reply when some background is developed.

---

### Author Response · Authors · 2020-11-18
**Result of additional experiments**

To clarify generallity of the proposed method for other types of biases as mentioned by reviewer 1 and reviewer 3, we performed additional experiments on three other biased data and the results were added to the Appendix of the paper. As a result, RMSE was reduced by 40% to 55% compared to the RMSE of the regression model trained with the skewed dataset and without the proposed approach. The other types of biased datasets includes a dataset with values within certain ranges were over sampled as mentioned by reviewer 1.

---

### Decision · Program_Chairs · 2021-01-07
**Final Decision**

**Decision:**

Reject

**Comment:**

This paper addresses the real-world problem of semi-supervised learning where the distribution from which the labeled examples are drawn is different from the distribution from which the unlabeled examples are drawn.  The task is motivated by structure-activity prediction for drug design (quantitative structure activity prediction, or QSAR).  Examples represent molecules, and we wish to predict a real-valued measure of binding affinity.  Exactly the general problem of data skew arose with exactly this task for example in one of the KDD Cup 2001 tasks.  While the authors here mention that labeled data may be focused more on active molecules (those with a high continuous-valued response), in the KDD Cup 200`1 data the reverse was true, and the unlabeled test data were skewed to higher activity level.  I say all this to agree with the authors about the real-world nature of the problem they address.  Also, some reviewers felt more empirical evaluation was needed, so that may be an additional data set for the authors to consider using.

Reviewer concerns including that the approach was simplistic, the empirical results were insufficient, and the claims were oversold.  The author replies and revisions, and the discussion, moved the reviews to be more favorable but still not strong enough to justify acceptance yet.  Nevertheless, the consensus is that the paper addresses an important problem and the revisions are headed in the right direction to make a strong future paper, and that the authors should be encouraged to continue this work.